# Sub-Scalp Implantable Telemetric EEG (SITE) for the Management of Neurological and Behavioral Disorders beyond Epilepsy

**DOI:** 10.3390/brainsci13081176

**Published:** 2023-08-08

**Authors:** Steven V. Pacia

**Affiliations:** Zucker School of Medicine at Hofstra-Northwell, Neurology Northwell Health, 611 Northern Blvd, Great Neck, New York, NY 11021, USA; spacia@northwell.edu

**Keywords:** SITE, sub-scalp, implantable EEG, telemetric-EEG, device, ambulatory EEG, sleep monitoring, neuro-behavioral disorders, degenerative brain disorders, Alzheimer’s disease, Parkinson’s disease, syncope

## Abstract

Sub-scalp Implantable Telemetric EEG (SITE) devices are under development for the treatment of epilepsy. However, beyond epilepsy, continuous EEG analysis could revolutionize the management of patients suffering from all types of brain disorders. This article reviews decades of foundational EEG research, collected from short-term routine EEG studies of common neurological and behavioral disorders, that may guide future SITE management and research. Established quantitative EEG methods, like spectral EEG power density calculation combined with state-of-the-art machine learning techniques applied to SITE data, can identify new EEG biomarkers of neurological disease. From distinguishing syncopal events from seizures to predicting the risk of dementia, SITE-derived EEG biomarkers can provide clinicians with real-time information about diagnosis, treatment response, and disease progression.

## 1. Introduction

SITE devices are fully or partially implantable, either subcutaneously or in subgaleal space. Developed for continuous ambulatory EEG monitoring over months to years, these devices have been designed to advance epilepsy management. Benefits include replacing frequently inaccurate patient diaries with objective seizure detection, [1] distinguishing focal from generalized seizures [2], identifying non-epileptic events, and monitoring antiseizure medication efficacy [3]. However, SITE devices will revolutionize the management of brain disorders beyond epilepsy. This article reviews the EEG biomarkers of various neurological disorders relevant to the limited channel SITE devices under development. Based on these foundational EEG studies, future applications for SITE are proposed.

## 2. Background and Rationale

SITE devices implanted fully or partially in sub-dermal or subgaleal space offer an opportunity for intermittent or continuous EEG monitoring over months to years. Scalp EEG is limited by the need for re-application and potential for skin breakdown that may occur after several days. Additionally, present-day intracranial devices are too invasive for most diagnostic EEG monitoring, especially if data may be collected with an extra-cranial device, implantable under local anesthesia in an outpatient setting. Although matching the resolution of 10–20 and higher density EEG is presently impractical with SITE, targeted chronic recordings over regions of interest, for example the vertex in sleep studies, may be very useful. Additionally, while intracranial EEG offers the ability to capture local, high-frequency discharges, unless performed with wide cortical coverage, data analysis may be “near-sighted”.

Outpatient and ambulatory EEG are used to inform diagnosis and treatment decisions almost exclusively for patients with epilepsy. EEG-guided clinical management, beyond epilepsy, is largely confined to clinical research, especially for neuro-behavioral and psychiatric patients. Most clinical research laboratories use quantitative EEG analysis to characterize baseline EEG data from patients with mood and attentional disorders. Statistical comparison of the data with controls can support clinical diagnoses and aid in drug selection [4]. Additionally, comparison of follow-up EEG data with baseline studies may aid in therapeutic monitoring and prognosis [5]. Studies are usually limited to brief supervised, mostly resting-state EEG sessions at pre-designated time intervals. Widespread use of EEG for neuro-behavioral disorder management has been limited by inconsistencies in protocols, discrepancies in findings between labs, limited expertise and time, and inconsistencies in reimbursement for the procedures [6]. Patient compliance with multiple recording sessions also hinders EEG-guided treatment strategies. The use of SITE devices, while sacrificing higher density EEG acquisition, offers several advantages to serial short-term EEG acquisition. SITE recordings allow EEG sampling at any time, without scheduled appointments, while awake and active or asleep. For therapeutic monitoring, unlimited sampling and comparison with baseline EEG is possible. Because electrodes are in fixed locations under the scalp, SITE recordings do not suffer from variations in electrode placement between recording sessions.

Most short-term EEG studies use international 10–20 electrode placements, with or without supplemental electrodes, for high-resolution EEG [7]. Higher density electrode array studies allow source localization, evoked responses, coherence analysis, and screening for focal dysfunctional brain regions. However, existing SITE systems designed for long-term continuous EEG or periodic sampling of EEG use limited electrode arrays, usually with one or two electrode pairs [2,3]. To maximize the diagnostic and therapeutic monitoring capability of SITE devices, baseline short-term EEG studies with full electrode arrays could guide SITE electrode placements to target specific lobar activity, resting-state rhythms like alpha, and state-dependent features like sleep spindles. Additionally, electrodes may be placed over homologous hemispheric regions for inter-hemispheric correlation studies [8].

A full review and recommendations for frequency and topographical analysis techniques of resting-state EEG were issued by the IFCN [9]. At present, the most useful analysis technique for the limited channel, SITE data, is the computation of the EEG amplitude/power density spectrum at specific electrodes and regions of interest. Amplitude or spectral density data may be calculated for pre-specified frequency bins or for overlapping frequency bands [10]. Statistical analysis of the quantitative data allows for comparison with baseline studies to identify hemispheric asymmetries and alterations in frequency band ratios that may be correlated with treatment response. Additionally, complex datasets with multiple variables may be analyzed with classifiers and machine learning (ML) algorithms to identify clusters of features correlated with diagnosis, prognosis, or treatment outcome. 

## 3. EEG Studies of Neurological and Behavioral Disorders

Table 1 lists potential EEG biomarkers relevant to SITE based on the following review of resting-state short-term EEG studies for several common neurological and neuro-behavioral disorders.

### 3.1. Depression

The conventional EEG reveals abnormalities in up to 40% of patients with depression [11]. EEG biomarkers can classify depression types and predict response to antidepressant therapy. One extensive review identified several classes of EEG biomarker, including band power, alpha asymmetry, evoked potentials, functional connectivity, and signal features [12]. Spectral EEG studies of band power analyzed with or without ML algorithms are by far the most prevalent. Although spectral EEG results vary due to technical differences in EEG acquisition and analysis [13], a few findings have been consistent and reproducible. Newson and Thigarajan reviewed 184 spectral EEG studies of psychiatric disorders, including depression [14]. The most consistent and frequent finding in depressed subjects was a 48% mean increase in the absolute theta and beta band power for both the eyes-open and eyes-closed resting states. A study of 1344 patients, using eLORETA, an EEG source localization technique, confirmed the finding of increased theta power across the frontal regions [15]. Additionally, several smaller studies highlight the importance of frontal alpha asymmetry, especially for predictions of prognosis and therapeutic monitoring [16,17]. Larger controlled studies are needed to confirm these assertions as well as claims of increased posterior cerebral theta power [18] and interhemispheric asymmetries of theta and delta band power in depressed patients [19].

More recently, gamma or high-frequency oscillations (HFOs), from 30 to 200 Hz, have been studied by both pre-clinical and clinical depression researchers. Gamma activity alterations distinguish monopolar from unipolar depression, separate depressed from healthy controls, and aid in predicting and monitoring therapeutic efficacy [20]. In a study of 533 depressed patients, high gamma power at various frontal, central, and temporal sites distinguished patients with suicidal ideation and suicide attempts from those without suicidal ideation [21]. This important finding awaits independent confirmation.

Increasingly, spectral EEG studies, especially those used to characterize antidepressant-induced EEG changes and their correlation with outcome, are leveraging ML algorithms to identify combinations of frequency and spatial characteristics predicting therapeutic response [22]. For example, ML methods identified a group of EEG biomarkers that predicted treatment response to SSRIs [4]. ML was also applied to wavelet analysis, where SSRI treatment responses were predicted with a high degree of sensitivity [23].

### 3.2. Attentional Disorders

To date, the most consistent EEG alteration associated with an attentional disorder is increased theta and decreased beta activity in the eyes-closed state of children with ADHD [14]. As a result, the theta/beta ratio (TBR) was approved by the FDA as a diagnostic biomarker of childhood ADHD. The spectral EEG data in adult ADHD has been less reliable, prompting one research group to question the diagnostic heterogeneity of adult subjects [24]. To standardize adult ADHD studies, Slater and colleagues recommend that future studies focus on ADHD subtyping by analyzing the resting-state and task-related modulation of alpha, beta, and theta power, together with event-related potentials. 

In addition to diagnosis, there is keen interest in the predictive ability of QEEG for ADHD outcome response to stimulants. Although a large prospective, multicenter open-label international study of 336 children and adolescents with ADHD compared with 158 healthy children failed to find TBR or alpha peak frequency group differences, male adolescents with lower frontal APF were less responsive to methylphenidate treatment [25]. The selectivity of the finding was attributed to maturational changes in the EEG, highlighting the need to focus research on well-selected age-matched populations. A subsequent smaller study of 51 subjects did positively correlate TBR, as well as delta/beta power, with methylphenidate treatment response [26]. Additionally, Arnett and colleagues, while confirming the predictive power of P3-amplitude-evoked responses for methylphenidate responsiveness, demonstrated distinct spectral EEG profiles for treatment responders and non-responders. Compared with responders, non-responders exhibited a flat, aperiodic, spectral slope in the frequency vs. power plot for the 1–50 Hz range [5]. In a separate spectral EEG study of 50 drug-naïve children receiving the SNRI medication atomexitine, theta cordance, a measure integrating absolute and relative band power, in the left tempo-parietal region predicted a positive treatment response at 6 weeks after only one week of treatment [27]. If reproducible by other researchers, EEG techniques like this could substantially impact ADHD treatment outcomes. 

### 3.3. Schizophrenia and Psychosis

In 2009, Galderisi and colleagues performed a comprehensive review of the available EEG studies of patients with schizophrenia [28]. For spectral EEG data, most studies revealed increased power in the theta and delta bands that occurred in both medicated and drug-naïve patients. A separate review of 37 studies confirmed the presence of absolute delta and theta band power increases as well as a reduction in absolute alpha power compared with controls in eyes-closed EEG studies [14]. The same was not found to be true in eyes-open EEG studies where absolute theta, alpha, and beta power were increased.

One study of patients with schizophrenia and their biological relatives confirmed the presence of increased front–central delta and central theta power in affected patients but not in their relatives, but they did find increased frontal beta power in both patients and their biologic relatives [29]. In contrast, bipolar disorder patients and their biologic relatives did not share similar spectral EEG characteristics.

A recent study of resting-state EEG gamma frequency power in 29 neuroleptic-naïve, non-affective first-episode psychosis patients revealed a significant increase in both the 31–50 and 51–70 Hz ranges compared with resting controls [30]. Additionally, shorter duration of illness prior to first psychotic episode was positively correlated with increased power in the 31–50 Hz gamma band in both frontal lobes. Further studies are needed to confirm the presence of increased frontal, lower end, gamma power at the time of first psychotic episode in schizophrenia. Perhaps, more importantly, studies determining the duration and predictive value of gamma activity and other EEG alterations prior to a first psychotic episode are needed. 

### 3.4. Obsessive Compulsive Disorder (OCD)

Compared with depression and ADHD, far less spectral EEG analysis has been applied to OCD. In a review of five spectral EEG studies in patients with OCD, like childhood ADHD and schizophrenia, the dominant abnormal EEG pattern was increased absolute and relative delta and theta band power [31,32]. Because OCD often co-exists with psychiatric disorders like ADHD and schizophrenia, caution in attributing the spectral EEG findings purely to OCD is advised [33,34].

### 3.5. Dementia

Diffuse slowing and disappearance of the posterior dominant alpha rhythm from a visual inspection of the EEG is well described in Alzheimer’s disease (AD) [35]. The degree of EEG slowing correlates with disease severity [36]. Spectral EEG studies confirm reductions in alpha and beta power combined with elevations in theta and delta band power [37]. Moreover, EEG abnormalities correlate well with the degree of cognitive impairment by neuropsychological testing and not only distinguish normal from those with dementia but predict which patients with minimal cognitive impairment (MCI) will progress to AD [38,39,40].

In addition, clinicians struggle to discern which patients with subjective memory complaints (SMCs) have true memory disorders. A study of patients with SMCs revealed significantly increased spectral theta density power and loss of central alpha reactivity with eye opening for older adults compared with younger subjects [41]. An elevated theta band power magnitude was positively correlated with degree of verbal memory loss. Similarly, an amyloid PET imaging investigation of cognitively normal adults with SMC separated patients into amyloid-positive and -negative groups [42]. The amyloid-positive PET subjects exhibited increased mid-frontal cortical theta power at the initiation of the study, but 24 months later showed increased theta in the posterior cingulate cortex and precuneus, thought to represent an anatomic progression of EEG abnormalities due to disease progression. 

Past dementia research, including spectral EEG studies, combined different types of dementia, leading to potentially misleading conclusions [43]. As a result, clinical characteristics, imaging, and pathology are being used to subtype patients for clinical research. EEG may provide an important physiological biomarker for dementia subtyping. Law and colleagues performed a systematic review of EEG studies, comparing patients with AD, DLB (dementia with Lewy bodies), and PDD (Parkinson’s disease dementia) [44]. Despite significant methodological differences between quantitative EEG studies, slowing of the dominant EEG rhythm (<8 Hz) assessed visually or through quantitative EEG was observed in nearly 90% of patients with DLB but only in 10% of patients with AD. In support of this finding, a separate study of 44 pathology-confirmed PDD patients concluded that delta band power positively correlated with Lewy body type synucleinopathy, especially in the anterior cingulate region, while the presence of AD or vascular pathology did not [45].

### 3.6. Parkinson’s Disease 

In addition to the study of PDD, EEG has been applied to the investigation of motor dysfunction in PD. A study of PD patients suffering from episodes of freezing of gait (FOG) demonstrated a significantly decreased power spectrum associated with voluntary stopping when compared with FOG episodes [46]. A distinct spectral pattern was observed in both delta and low-beta power in central brain regions. EEG may also provide biomarkers of PD severity. By acquiring both resting-state EEG and dopamine transporter PET imaging in PD patients off medication, Waninger and colleagues correlated disease severity and reduced dopamine transporter activity with excess EEG beta coherence [47].

### 3.7. Obstructive Sleep Apnea (OSA)

Spectral EEG studies of OSA, performed during polysomnogram, show enhanced beta band power during NREM sleep and increased delta power in either REM or NREM sleep [48,49]. Additionally, delta band power spectral density correlates positively with the apnea hypopnea index, longest time of apnea, oxygen desaturation index, and percent sleep time below 90% SaO_2_ [50]. Conversely, the arousal index correlates negatively with slow wave sleep percentage and minimum oxygen saturation. Additionally, spectral EEG beta band power appears to decrease with lessening OSA severity [51].

### 3.8. Syncope

The lifetime cumulative incidence of syncope in the general population is as high as 35% [52]. Vasovagal or neuro-cardiogenic syncope is the most frequent etiology for all patients, with cardiac syncope increasing in incidence with age [53]. EEG, performed during syncope induced by tilt table testing, showed generalized theta and delta slowing in all patients, with many progressing to diffuse background suppression or a “flat EEG.” [54]. Early theta and delta slowing may also be seen during the pre-syncopal phase [55]. Regardless of etiology, syncope due to cerebral hypoperfusion causes generalized EEG slowing, progressing to background attenuation if ischemia continues [56].

Distinguishing syncope from seizure remains a diagnostic challenge, often requiring extensive neurological and cardiac evaluations. This is especially true for patients presenting with convulsive syncope [57]. Moreover, given the sporadic nature of syncopal events, capturing episodes with short-term ambulatory scalp EEG occurs very infrequently. Given the sensitivity of EEG to distinguish syncope from seizures, SITE is an excellent diagnostic option for patients with recurrent unexplained loss of consciousness. 

### 3.9. Brain Tumor

The standard of care for managing potentially progressive brain tumors, like gliomas, is serial MRI scanning. Unfortunately, even when performed at relatively short intervals, usually 3–6 months, tumor progression may be identified in patients without new neurological symptoms, delaying potentially life-saving therapy. Preliminary studies indicate that analyzing multiple spectral EEG features reliably discriminates a normal brain from a brain infiltrated with tumor [58]. Of course, serial resting EEG follow-up for brain tumors would suffer from the same diagnostic delay as serial MRI. However, SITE has the potential to identify local EEG alterations related to tumor growth or swelling, in real time, before new symptoms occur. Additionally, the magnitude of source theta and gamma band power has been correlated with glioma grade [59]. If EEG background alterations occur during malignant transformation of low-grade tumors, then SITE devices could provide the early detection of tumor progression. Finally, high-frequency oscillations (HFOs) have been associated with seizure onset location in peritumoral regions [60]. Continuous SITE studies aimed at identifying the development of HFOs could potentially predict future seizure risk or tumor progression. 

### 3.10. Traumatic Brain Injury (TBI)

EEG performed early on after moderate-to-severe head trauma predicts neurological outcome [61,62]. Using ML techniques, a set of EEG biomarkers was identified that correlated well with clinical outcome at six months [61]. Similarly, a study of TBI patients with continuous ICU EEG, using a random forest classifier to identify EEG features at 72 h, predicted poor clinical outcome at 12 months with reasonable accuracy [62]. A separate study aimed at quantifying the effects of repeated lower-level head traumas derived a quantitative EEG algorithm, through ML, that could distinguish former NFL players from age-matched controls. Additionally, the EEG algorithm distinguished players who began playing as adolescents from those who started later in life [63]. The EEG features were comparable to those seen in neurodegenerative disorders.

EEG background changes soon after trauma also predict the development of post-traumatic epilepsy (PTE) [64]. In 63 PTE patients, those with more EEG background suppression and beta variability on admission had a greater chance of developing PTE. 

The progression of EEG changes in the months following TBI has not been well studied. Considering the predictive value of early EEG in TBI, SITE devices may play a role in monitoring the efficacy of neuro-rehabilitation interventions while also identifying seizures in these vulnerable patients.

**Table 1 brainsci-13-01176-t001:** EEG biomarkers of neurological disease and therapeutic response applicable to SITE devices.

Disorder	EEG Biomarkers *
Depression	Absolute theta and beta band power elevation [14] Frontal theta band power elevation [15] HFOs [20,21] ML features [4,22,23]
Attentional Disorders	TBR elevation [14] APF reduction [25]
Schizophrenia	Absolute delta and theta band power elevation [28] Absolute alpha band power reduction [14] Gamma band power elevation [30]
OCD	Absolute and relative delta and theta band power elevation [31,32]
Dementia (including AD and PD)	Alpha and beta band power reduction [37] Theta and delta band power elevation [37]
PD (motor dysfunction)	Central spectral delta and low-beta band power elevation (FOG) [46] Beta coherence excess [47]
OSA	Beta and delta band power elevation [48,49]
Syncope	Theta and delta slowing [55] Background suppression [54]
Brain Tumor	Theta and gamma power elevation [59]
TBI	ML features [61,62]

* Based on resting-state routine scalp EEG studies, OCD—obsessive compulsive disorder, AD—Alzheimer’s disease, PD—Parkinson’s disease, OSA—obstructive sleep apnea, HFO—high-frequency oscillations, ML—machine learning, TBR—theta/beta ratio, APF—alpha peak frequency, FOG—freezing of gait.

## 4. SITE for the Management of Neurological Disorders 

### 4.1. SITE Device and Electrode Implantation

Minimally invasive SITE electrodes are implanted into the subdermal or subgaleal spaces over the cranium [65]. Electrodes may be inserted in a lateral to medial direction toward the vertex or from posterolateral to anterolateral to cover the temporal region [2,3]. In either case, electrode number and spatial resolution are limited to ensure minimally invasive procedures and long-term patient comfort [66,67]. Subgaleal device and electrode placement between the temporalis muscle insertions may result in less muscle artifact than laterally placed electrodes under the temporalis muscles [2,65]. Figure 1 shows a coronal MRI of a six-contact, subgaleal EEG electrode inserted above the left temporalis muscle and directed toward the cranial vertex. For patients where bi-hemispheric EEG sampling is necessary, the distal electrode may be inserted over the vertex contralaterally, leaving the proximal electrode ipsilateral, near the incision site. Alternatively, bi-hemispheric EEG sampling may be accomplished through bilateral device implantation [68].

Fully implantable SITE systems may be preferable to those with external batteries, eliminating the need for patient compliance [65]. Depending on the desired amount of EEG sampling and wireless signal transmission, implanted batteries are expected to last between two and three years before requiring replacement. SITE devices that use an external battery, similar to cochlear implants, have rechargeable batteries but require ongoing patient compliance to ensure continuous EEG sampling.

The operative procedure for fully implantable SITE devices may be performed in the outpatient setting under local anesthesia. Following a single scalp incision, a ribbon retractor shaped to approximate the curvature of the skull may be used to bluntly dissect a subgaleal pocket between the galea and the pericranium. Because the subgaleal space is a potential space, the body of the device and electrode array may be inserted into the pocket without concern for migration [2]. For partially implantable devices, only the electrode and transmitter need to be implanted, resulting in a smaller subgaleal dissection track [3]. However, maintaining proximity to the post-auricular external power source requires electrode insertion to pass under the temporalis muscle. Fully implantable SITE systems, as shown in Figure 2, may be deployed between the insertions of the temporalis muscles, over the cranium, essentially eliminating temporalis and mandibular muscle artifact.

### 4.2. Data Collection and Analysis

SITE device data will corroborate and extend our understanding of the neurophysiology of neurological disease, derived from resting-state, short-term EEG studies. Conversely, ongoing routine quantitative EEG studies, using high-density electrode arrays, can direct SITE electrode placement locations, guide analysis methods, and maximize the value of data acquired from lower spatial resolution SITE devices. Although acquisition and analysis methods applied to SITE are potentially limitless, below we consider preliminary data strategies that may guide long-term EEG management of neurological and behavioral disorders. 

Most quantitative EEG techniques, including power spectral density analysis, may be applied to SITE. Data are collected similarly to serial resting-state routine EEG, except awake and sleep EEG epochs are either pre-scheduled or triggered, using a classifier that detects targeted signal features that will initiate the transmission and storage of EEG for visualization and analysis [69]. Continuous data acquisition is possible but reduces battery life due to frequent data transmission. Additionally, acquiring and analyzing continuous data from a device over weeks, months, or years is challenging from a transmission, storage, and computational standpoint. Fortunately, data classifiers extracting and sorting features of complex data signals, like EEG, are being developed for epilepsy monitoring, sleep staging, and ICU surveillance, reducing the amount of data transferred [69,70,71]. For example, a simple automated seizure detection algorithm, based on an expert EEG review, used spectral density and line length to detect seizures from study participants [69]. The seizure detection paradigm performed with greater than 90% accuracy. Similar paradigms may be devised to detect and store sleep complexes, rhythmic slowing, and HFOs, features relevant to the study of non-epilepsy brain disorders. These designated EEG epochs can be transmitted to a smart phone and then to the cloud for expert review and quantitative off-line analyses.

For the study of complex, multi-modal data, ML techniques can automate the review of large complex datasets, like polysomnograms or critical care data. Chaudhry and colleagues reviewed both ML and standard statistical approaches (SSAs) for the analysis of the complex neurological and physiological data generated in neuro-critical care settings [71]. They concluded that SSAs are hypothesis-driven, while ML is not. SSAs target linear data, while ML may be applied to multi-dimensional, non-linear data. Finally, ML algorithms are better suited for predictions than SSAs. The utility and power of ML was highlighted by a recent study of sleep EEG biomarkers [70]. By correlating EEG changes with sleep-stage-related alterations in the polysomnogram, sleep-stage-specific EEG biomarkers were identified, allowing sleep staging to be assessed by EEG alone. 

### 4.3. Therapeutic Monitoring

EEG monitoring with SITE devices presents an unprecedented opportunity for therapeutic monitoring. Continuous EEG screening and automated cloud-based analysis provide real-time access to therapeutic responses, not possible with routine laboratory-based EEG. For example, after establishing baseline TBR or alpha peak frequency for an ADHD patient, the effects of medication or cognitive behavioral therapy on the EEG may be assessed immediately, allowing for timely provider intervention. Similarly, instead of repeated polysomnograms or following oxygen saturation to assess the efficacy of OSA treatments, SITE EEG may provide both nightly sleep staging as well as verification of improved waking cerebral function and alertness, which is the ultimate goal of OSA treatment. 

### 4.4. Overcoming Spatial Sampling Limitations

Due to the lower spatial resolution of SITE systems, the complex modeling of cortical sources and cerebral networks that require high-density electrode arrays are less appropriate for SITE devices [9]. Similarly, EEG microstate analysis which attempts to define very brief 80 to 100 msec stable global patterns of scalp potential topographies used to characterize large-scale neural networks would also be challenging for limited channel electrode arrays [72]. However, resting-state studies often lead to the discovery of limited cortical locations or regions of interest that characterize disease progression or therapeutic response. For example, short-term EEG studies using standard 10–20 electrode placements, for hemispheric asymmetry analysis, frequently identify asymmetries in single electrode pairs, like P3–P4 [73]. Targeted SITE electrode implantation, based on well-conceived based-line, high-density EEG studies, obviates the need for whole-brain EEG coverage, allowing for efficient data acquisition and focused analysis.

### 4.5. Sleep EEG Analysis

Sleep offers a structured and consistent opportunity to follow brain function over time. Discrete sleep oscillations, like sleep spindles that characterize stage 2, NREM sleep, are associated with memory consolidation [74,75]. Moreover, alterations in sleep spindle frequency and architecture occur in AD [76], schizophrenia, Autism [77], and PD [78]. For example, a study of sixty-eight PD patients without dementia were compared with forty- seven healthy controls to determine if sleep spindle analysis predicts dementia [78]. Sleep spindle density and amplitude were reduced in the eighteen patients who later developed dementia compared with the controls and PD patients who did not develop memory disorders. In addition to sleep spindle analysis, spectral density and topography features of synchronous low-frequency NREM sleep oscillations correlate with prognosis for memory and cognitive disorders, including AD [79].

More recently, closer analysis of sleep spindles shows significant intra-subject heterogeneity, leading to new analysis strategies to integrate continuous variables like frequency, amplitude, duration, and topography to identify patient-unique sleep biomarkers [80]. For example, instead of averaging spindle activity across sleep stages, Stokes and colleagues considered the complex temporal evolution of transient events, like spindles, as a dynamic process and developed a continuous sleep metric that proved to have strong night-to-night stability for individuals [81]. By coupling slow oscillation power with phase to create a power-phase representation for a full-night sleep recording, it becomes possible to create an EEG phenotype for an individual. The technique was applied to seventeen patients with schizophrenia, confirming previously established sleep spindle deficits but also discovering unique transient NREM events in low alpha and theta ranges that characterized each patient’s sleep. By applying dynamic analysis techniques like these to SITE, sleep biomarkers may reflect therapeutic response and disease progression not only for schizophrenia but for all cognitive disorders. Additionally, because sleep complexes and spindles are seen at the mid-line, SITE electrodes placed at the cranial vertex are appropriate for sleep EEG. 

## 5. Conclusions

The clinical practice of medicine is evolving rapidly due to the availability of implantable medical devices that provide the physician with continuous biometric data, from blood glucose concentrations to cardiac rhythm surveillance and more. Presently, however, few devices are available for the continuous monitoring of brain physiology. SITE devices under development for seizure detection may not only improve epilepsy management but could extend our understanding of EEG biomarkers that characterize all other brain disorders. Analysis of continuous EEG with advanced techniques such as ML has the potential to improve neurological diagnosis and management, advance therapeutic monitoring, and usher in a new era of personalized neurological care.

## Figures and Tables

**Figure 1 brainsci-13-01176-f001:**
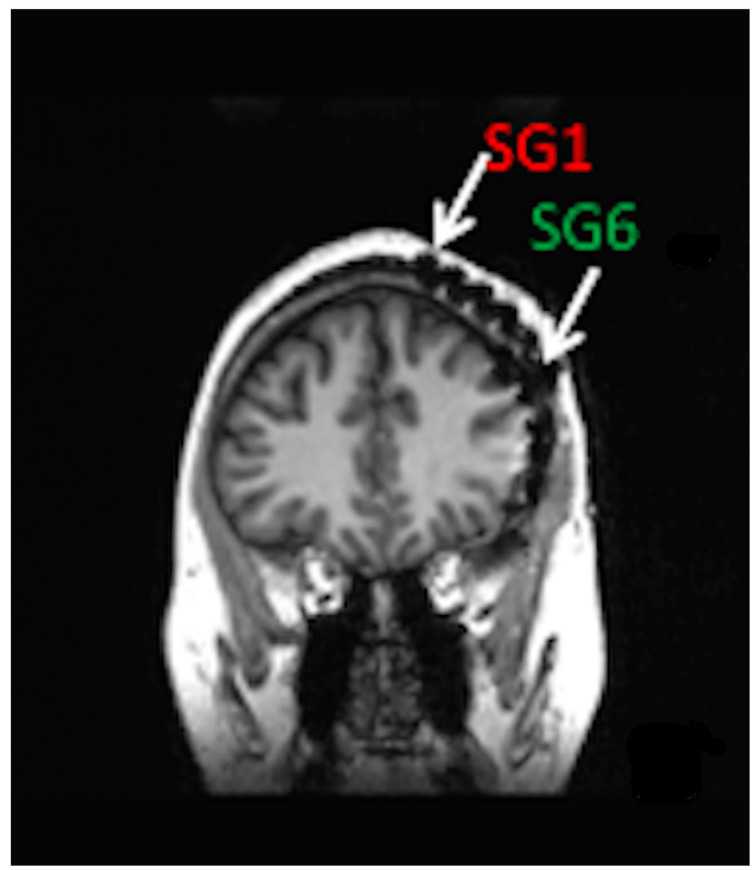
Coronal MRI of patient with left hemisphere intracranial EEG electrode placement with simultaneously placed 6-contact, subgaleal EEG electrode with the most distal contact, SG1 near the vertex and most proximal contact, and SG6 laterally at the site of electrode placement. For bi-hemispheric sampling, SG1 would be directed contralaterally past the vertex, with SG6 remaining over the left hemisphere.

**Figure 2 brainsci-13-01176-f002:**
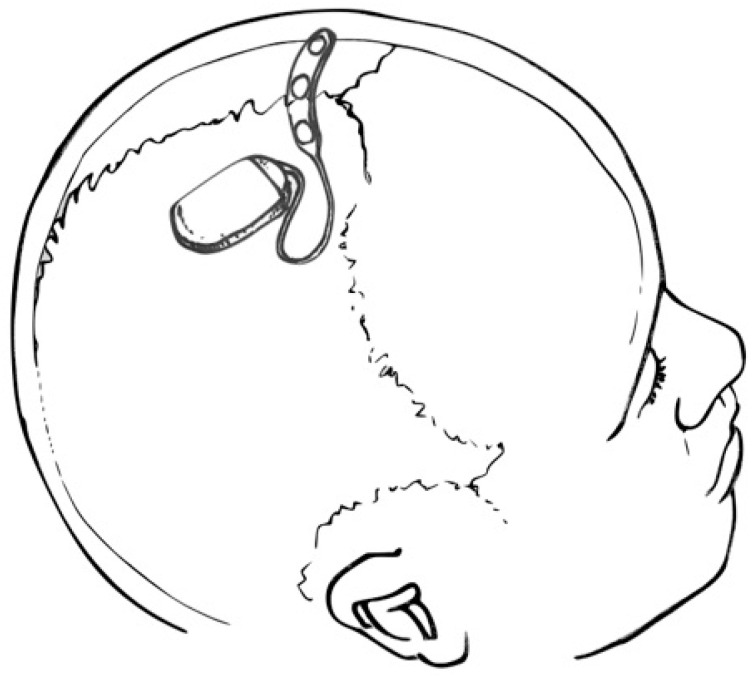
Drawing depicting a SITE device and multi-contact strip electrode placed over the cranial vertex (within the subgaleal space—not shown). Courtesy of artist/surgeon Werner K. Doyle, MD.

## Data Availability

Not applicable.

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
