# Peer review of "Sub-Scalp Implantable Telemetric EEG (SITE) for the Management of Neurological and Behavioral Disorders beyond Epilepsy"

_brainsci, 2023, doi:10.3390/brainsci13081176_

Round 1
Reviewer 1 Report
Monitoring of the functional state of the brain is indeed an important and one of the key methods for diagnosing epilepsy. Recently, however, interest in this method has been growing, and the area of diseases for which this method can be applied is expanding. Currently, both scalp electroencephalographic methods and cortical intraoperative methods are used. Preoperative preparation of patients with epilepsy includes multi-day electroencephalographic monitoring using, among other things, immersed electrodes for recording ictal events. However, these studies are carried out strictly in stationary conditions. Therefore, the invasive Sub-scalp Implantable Telemetric EEG (SITE) technique discussed in the article, which probably allows multi-day monitoring, especially on an outpatient basis, is very relevant.
However, a number of questions arose during the review process:
1) Authors are encouraged to structure the abstract in subsections (according to the sections of the manuscript);
2) The text of the article describes the application of electrodes by the SITE method, however, for greater clarity, it is necessary to provide drawings with a schematic arrangement of electrodes (or photographs of patients);
3) The SITE technique is not described in detail, the authors need to add a separate section technically describing the technique, including the electrode implantation process. It is also necessary to add a comparative description of the SITE with existing similar methods, their pros and cons;
4) There is no indication in the manuscript for how long the electrodes can be implanted, whether video surveillance of the patient was carried out, if not, how motion artifacts were excluded;
5) The quality of picture 1 does not match, it is recommended to replace picture 1 with a higher quality picture.
Author Response
Dear Reviewer,
Thank you for taking the time to review this article. I hope that I have addressed your concerns.
Reviewer 1
1) Authors are encouraged to structure the abstract in subsections (according to the sections of the manuscript);
Your editor has helped with this-thank you
2) The text of the article describes the application of electrodes by the SITE method, however, for greater clarity, it is necessary to provide drawings with a schematic arrangement of electrodes (or photographs of patients);
I have included a drawing, new figure 2, of the device and electrode strip and how it is situated on the skull.
3) The SITE technique is not described in detail, the authors need to add a separate section technically describing the technique, including the electrode implantation process. It is also necessary to add a comparative description of the SITE with existing similar methods, their pros and cons;
I have added a paragraph in section 2 to introduce these devices.
I have also added two new paragraphs to section 4.1 that discuss the implantation process and compares the two types of SITE devices- fully implantable and partially implantable.
4) There is no indication in the manuscript for how long the electrodes can be implanted, whether video surveillance of the patient was carried out, if not, how motion artifacts were excluded;
I have added a statement in section 2- months to years as well as a sentence in section 4.1 discussing battery life. I have discussed muscle artifact with devices placed below and above the temporalis muscle insertions. Videos were done in my paper in JCN for epilepsy patients but not planned as part of any devices under development (yet!)
5) The quality of picture 1 does not match, it is recommended to replace picture 1 with a higher quality picture.
I have emailed a higher resolution MRI picture to your editor
All the best,
Steve Pacia
Reviewer 2 Report
The paper investigates the possibilities of using wub-scalp implantable telemetric EEG to manage neurological and behavioural disorders.
The paper is well structured, with appropriate references and detailed sections on neurological and behavioural disorders.
Author Response
Thank you for taking the time to review this article.
Much appreciated,
Steve
Reviewer 3 Report
This is a well-written review that covers an interesting topic. The use of EEG beyond epilepsy is an area that is under appreciated. This review provides some fresh perspectives.
I have only one minor concern. Personally, I believe that section 4.1-4.2 should be moved to the beginning of the review after the introduction. The current section 2 is difficult to follow in the absence of a basic description of the SITE.
Author Response
Thank you for the taking time out to review this paper. I agree with your concern. I hope this suffices.
Steve
Reviewer 3- Because of the added detail about the operation and different SITE devices requested by reviewer 1, it became too detailed for the beginning of the paper so I wrote a new introductory paragraph to section 2 so that the reader has a better idea of SITE before the EEG data review begins.